# Polypharmacy Management in Chronic Conditions: A Systematic Literature Review of Italian Interventions

**DOI:** 10.3390/jcm13123529

**Published:** 2024-06-17

**Authors:** Lara Perrella, Sara Mucherino, Manuela Casula, Maddalena Illario, Valentina Orlando, Enrica Menditto

**Affiliations:** 1CIRFF-Center of Pharmacoeconomics and Drug Utilization, Department of Pharmacy, University of Naples Federico II, 80131 Naples, Italy; lara.perrella@unina.it (L.P.); sara.mucherino@unina.it (S.M.); valentina.orlando@unina.it (V.O.); 2Epidemiology and Preventive Pharmacology Service (SEFAP), Department of Pharmacological and Biomolecular Sciences, University of Milan, 20133 Milan, Italy; manuela.casula@unimi.it; 3IRCCS MultiMedica Hospital, Sesto S. Giovanni, 20099 Milan, Italy; 4Division of Health Innovation, Campania Region Health Directorate, 80143 Naples, Italy; illariomaddalena@gmail.com

**Keywords:** polypharmacy, multimorbidity, adult population, chronic conditions, management, inappropriate polypharmacy, interventions, healthcare costs

## Abstract

**Background:** Potentially inappropriate polypharmacy (PIP) is among the major factors leading to adverse drug reactions, increased healthcare costs, reduced medication adherence, and worsened patient conditions. This study aims to identify existing interventions implemented to monitor and manage polypharmacy in the Italian setting. **Methods:** A systematic literature review (PROSPERO: CRD42023457049) was carried out according to the PRISMA statement guidelines. PubMed, Embase, ProQuest, and Web of Science were queried without temporal constraints, encompassing all published papers until October 2023. Inclusion criteria followed the PICO model: patients with polypharmacy; interventions to monitor/manage polypharmacy regimen versus no/any intervention; outcomes in terms of intervention effectiveness and cost variation. **Results:** After duplicate deletion, 153 potentially relevant publications were extracted. Following abstract and full-text screenings, nine articles met the inclusion criteria. Overall, 78% (*n* = 7) were observational studies, 11% (*n* = 1) were experimental studies, and 11% (*n* = 1) were two-phase studies. A total of 44% (*n* = 4) of the studies involved patients aged ≥ 65 years, while 56% (*n* = 5) were disease-specific. Monitoring was the most prevalent choice of intervention (67%; *n* = 6). Outcomes were mainly related to levels of polypharmacy (29%; *n* = 6) and comorbidities (29%; *n* = 6), effectiveness rates (14%; *n* = 3), and avoidable costs (9%; *n* = 2). **Conclusions:** This review outlines that Italy is still lacking in interventions to monitor/manage PIP, addressing an unmet need in developing patient-tailored strategies for reducing health-system burden.

## 1. Introduction

Multimorbidity, as per the World Health Organization’s characterization, refers to the simultaneous presence of two or more chronic medical conditions within an individual [1]. Patients with multimorbidity may require medicines to treat each condition, which can lead to polypharmacy. Currently, around 50 million EU citizens are estimated to have multimorbidity. However, polypharmacy, defined as prescriptions for from five to ten medications [2], is not always appropriate. Inappropriate polypharmacy, characterized by the excessive or inappropriate use of multiple medications, is not only associated with adverse clinical outcomes, but can lead to rising costs for both patients and the healthcare system, contributing to the wastage of scarce health resources [3].

Notably, polypharmacy occurs in more than 1.3 million individuals; in particular, the estimated prevalence in Italy stands at 32.9% [4,5]. The escalating incidence of inappropriate polypharmacy underscores the critical need for a nuanced understanding of prescribing practices and their potential adverse effects within the context of complex healthcare regimens [6]. Hence, several studies have shown that inappropriate polypharmacy increases the risk of unnecessary drug use, potential drug–drug and drug–disease interactions, and adverse drug reactions (ADRs) [7,8,9]. The complexity of managing polypharmacy arises from the intricate interactions and potential contraindications among multiple medications, necessitating careful consideration to avoid adverse effects. Therefore, inappropriate polypharmacy management involves complex decision making and requires the combined knowledge of physicians, pharmacists, and nurses supported by informed patient interaction [6]. The complexity and heterogeneity of patients with multimorbidity and polypharmacy renders traditional disease-oriented guidelines often inadequate and complicates clinical decision making. There is a clear need to prioritize research to generate evidence for effective interventions in ‘real-world patients’ [10]. Hence, Kurczewska-Michalak M, et al. (2021) have already analyzed the European context through a scoping review. It was found that multiple approaches toward polypharmacy management are advised in the current literature [11].

There is a variety of tools aimed at reducing inappropriate polypharmacy using either implicit (judgement-based) or explicit (item list-based) criteria [12,13,14]. Unfortunately, the implementation in real practice of these approaches is very limited because of their varying replicability, complexity, and applicability [11,12,13,14]. Currently, no gold standard exists for polypharmacy management, and various approaches are used in the European context [11,12,13,14]. A peculiar case in point is Italy, which is among the top five European countries with the highest prevalence of polypharmacy [4]. Hence, the purpose of this study was to analyze Italy as a peculiar case in the European context to identify how inappropriate polypharmacy is managed in the real world and to provide a benchmark in Europe.

## 2. Materials and Methods

### 2.1. Search Strategy 

A systematic review of the literature was conducted according to the PRISMA (Preferred Reporting Items for Systematic reviews and Meta-Analyses) statement guidelines [15]. This study was registered with the international database of prospectively registered systematic reviews PROSPERO (ID number: CRD42023457049). 

A systematic literature search was performed in PubMed/MEDLINE, EMBASE, ProQuest, and Web of Science for articles published until October 2023. In addition to electronic searches, the references of all included articles were crosschecked. The search strategy combined headings and keywords used to define the target Population, Intervention, Comparator, and Outcome (PICO), searched as MeSH (Medical Subject Headings) or Emtree terms and in the title and abstract. The strategy combined six major themes with their synonyms: (i) polypharmacy; (ii) management; (iii) adult population; (iv) any type of intervention; (v) healthcare costs; (vi) Italian setting. Boolean operators AND/OR were employed. 

Appendix A provides the combination of search terms that were used to identify relevant publications. The search strategy is summarized in Appendix A showing the number of records obtained per literature database.

### 2.2. Eligibility Criteria 

All peer-reviewed original articles about existing interventions apt to monitor and manage PIP in the Italian setting, published until October 2023 and responding to the PICO, were considered for inclusion in the study. Studies conducted in different countries that included Italy were included if they showed clear results concerning Italy. On the other hand, conference proceedings, rationale and/or study protocols, letters, editorials, commentaries, reviews, consensus, guidelines, expert opinions, and gray literature were not included. All studies conducted in countries other than Italy were excluded. Moreover, language restriction was applied to the search, as fundamental to the eligibility of the study was the availability of the papers’ full text published in English and Italian. Any identified literature reviews were used as a source for finding additional articles not present in our data set. The inclusion and exclusion criteria are summarized in Table 1. 

### 2.3. Selection and Data Processing

Study selection (title/abstract screening and full-text screening) was performed by two reviewers (L.P. and S.M.) independently. Any disagreements between reviewers were discussed until a consensus was reached by involving two other reviewers (E.M. and V.O.). All references were screened for relevance and those potentially eligible were assessed according to the inclusion/exclusion criteria and accepted or rejected, as appropriate. Titles and abstracts were screened to discard irrelevant papers in the first screening phase. Then, full texts of the records selected from the previous step were retrieved and screened to assess their eligibility for inclusion in the qualitative analysis. The type of information that was extracted from each reference included in the qualitative analysis and collected into a dedicated file is reported in Table 2. 

### 2.4. Data Synthesis 

Information was collected separately for observational and experimental studies. Authors, year, population age, number of patients, diagnosis, intervention type, study outcomes, outcome measures, and tool used were extracted from each observational study. Authors, year, study design, population age, number of patients, intervention type, tool used, intervention setting and healthcare professionals (HCPs) involved, study outcomes, and outcome measures were extracted from each experimental study.

### 2.5. Quality Appraisal

The quality of studies was assessed separately according to the study design. The quality of observational studies was assessed according to the STROBE (STrengthening the Reporting of OBservational studies in Epidemiology) Statement Checklist [16] shown in Appendix A. The experimental studies’ quality appraisal was assessed using the CASP (Critical Appraisal Skills Programme) Randomised Controlled Trial Standard Checklist [17] shown in Appendix A, assessing heterogeneity, consistency, and risk of bias. On the one hand, the STROBE checklist contains a total of 22 items which evaluate the reporting of each study’s title, abstract, introduction, methodology, results, and discussion. On the other hand, the CASP checklist contains a total of 11 screening questions, divided into Section A about the validity of the basic study design, Section B to assess whether the study was methodologically sound, and Sections C and D about the robustness of the results. Quality was evaluated according to each of the STROBE items and CASP sections, indicating their presence or absence in the selected studies. The Completeness of Reporting (COR) score for each manuscript was calculated as the “yes” answers as a proportion of the “yes + no” answers: COR score (%) = [yes/(yes + no)] × 100 [18]. Quality was measured according to previous studies: that is, “low” (COR: 0–49%), “moderate” (COR: 50–74%), and “high” (if ≥75% of items were met) [19,20]. 

## 3. Results

### 3.1. Overview of Included Studies 

The search strategy produced 154 potentially relevant publications, of which 89 papers (17 duplicates and 73 by publication type) were removed. After title and abstract screening of the remaining 64 records, 14 potentially relevant publications were retained according to the inclusion criteria. After a full-text review, five articles were excluded according to the exclusion criteria. The reasons for exclusion are listed in the diagram (Figure 1). Thus, nine articles were included in the qualitative evaluation. 

The nine included studies were divided according to their study design, as follows:Observational studies assessing intervention to manage/monitor PIP and/or associated avoidable costs (*n* = 7) [21,22,23,24,25,26,27];Experimental studies carried out on tailored interventions to manage/monitor PIP (*n* = 1, two-phase–observational [28]; *n* = 1, experimental study [29]).

### 3.2. Risk of Bias within Studies

The risk of bias is presented for each study in Appendix A. The seven observational studies included in the systematic review were generally of high quality as rated by STROBE criteria, complying with a score between 80 and 90% for *n* = 4 studies [21,23,25,26], and a score above 90% for *n* = 3 studies [22,24,27] (Appendix A). The two experimental studies included in the systematic review were both of high quality as rated by CASP criteria, both complying with a score of 75% (Appendix A) [28,29].

### 3.3. Observational Studies

Seven (78%) observational studies were identified, conducted from 2016 to 2021. Of these studies, three considered the entire Italian territory, one study assessed centers across northern and central Italy, one considered the population of Lombardy [22], and two studies were conducted in Veneto (one in Padua [25] and one in Verona [26]), represented in Figure 2. 

Table 3 describes the characteristics of these studies. Overall, 29% (*n* = 6) of the studies reported levels of polypharmacy [21,22,23,24,25,26], 29% (*n* = 6) reported levels of comorbidities [21,22,23,25,26,27], and the remaining 43% reported effectiveness rates (*n* = 3) [21,24,27], costs (*n* = 2) [26,27], drug consumption rates [23], the proportion of potentially inappropriate medications (PIMs) [23] and of drug–drug interactions (DDIs) [25], and the level of medication adherence [27]. 

In all cases, the target population consisted of adult patients over 18 years (*n* = 3) [21,22,26], older patients over 65 years (*n* = 2) [24,25], adult patients over 35 years (*n* = 1) [23], and adult patients over 40 years (*n* = 1) [27].

Patient complexity was considered in all studies, whose main outcomes are detailed in Appendix A. The mean age observed ranged from 63 to 78 years. Levels of comorbidity were considered using the Charlson comorbidity index [21], multisource comorbidity score [22], and multidimensional prognostic index (MPI) [25] or calculated in terms of prevalence (n%) [26,27]. The majority of the studies showed that groups of patients with a higher mean age also had a higher prevalence of comorbidities. Furthermore, it was found that healthcare services utilization was higher in older than in younger age groups. The average number of prescriptions was more than eight times higher in the ≥81 age groups compared with the youngest groups [23]. Therefore, in a sample with a median age of 78 years, a Charlson comorbidity index ≥ 3 showed the highest prevalence (*n* = 374; 31.2%) [21]. In contrast, a sample with a median age of 63.1 years showed a higher prevalence of comorbidities scored 0–1 (*n* = 71) than comorbidities scored ≥2 (*n* = 28) [27].

The most frequent comorbid illnesses and prescribed medications were found to be the cardiovascular ones [21,22,24,25]. All main outcomes are detailed in Appendix A. 

Fifty-six percent (*n* = 5) of the studies were disease-specific, three of which considered respiratory diseases, such as pneumonia, asthma, and COPD [22,26,27], while the other two considered LGIB (lower gastrointestinal bleeding) [21] and HIV (human immunodeficiency virus) [24].

Among all of the observational studies, 86% (*n* = 6) used a monitoring intervention [21,22,23,24,26,27] and 14% used a decision support tool as intervention, proposing tools such as STOPP (screening tool of older persons’ potentially inappropriate prescriptions) criteria and the Micromedex^TM^ Drug-Reax System (*n* = 1) [25].

Out of all of the studies conducted, only two considered costs as one of the calculated outcomes [26,27]. Direct and indirect costs per patient per year were considered. Both studies showed a decrease in total costs after intervention.

### 3.4. Experimental Studies

Two (22%) experimental studies were identified, conducted from 2008 to 2013; both were conducted in the Lombardy region (Figure 2). Table 4 describes the characteristics of these studies. The target population consisted of older patients over 65 years in both studies (*n* = 2) [28,29]. Both samples showed a prevalence of women, with a mean age of 81.3 and 82.3 years.

Changes in parameters pre- and post-intervention were explored, whose main outcomes are detailed in Appendix A.

Ghibelli S, et al. investigated the level of PIMs, DDIs, comorbidities, drug consumption, and effectiveness rates in terms of prevalence. In the intervention phase versus the observation phase, a higher proportion of patients with PIMs (*n* = 25; 41.7% vs. *n* = 29; 39.1%) and DDIs (*n* = 51; 85.0% vs. *n* = 64; 86.5%) were recognized at hospital admission, and a significantly lower number at discharge for both PIMs (*n* = 7; 11.6% vs. *n* = 28; 37.8%) and DDIs (*n* = 53; 88.3% vs. *n* = 65; 87.8%) [28]. 

Moreover, Onder G, et al. conducted an experimental study proposing a homecare program based on case management in several European countries, including Italy. This study showed that the caregivers of subjects in case manager groups were less likely to be unable to continue undertaking caring activities (*n* = 49/1320; 4% vs. *n* = 134/2129; 6%, *p* < 0.01) and were less dissatisfied (*n* = 28/1320; 2% vs. *n* = 83/1129; 4%, *p* < 0.001) compared with those in the no case manager group [29]. 

**Table 3 jcm-13-03529-t003:** Characteristics of included observational studies.

Authors	Year	Population Age	Geographical Area	Diagnosis	Intervention Type	Study Outcomes	Outcome Measures	Polypharmacy	Tool Used
Radaelli F, et al. [21]	2021	18+	The whole of Italy	LGIB	Monitoring	-Epidemiology rates -Drug consumption rates -Clinical outcomes	-Level of polypharmacy (%) -Level of comorbidities (n%) -Effectiveness rates (n%)	Secondary outcome	NS
Faverio P, et al. [22]	2020	18+	The whole of Italy	Pneumonia	Monitoring	-Epidemiology rates -Drug consumption rates	-Level of polypharmacy (n%) -Level of comorbidities (n%)	Secondary outcome	NS
Atella V, et al. [23]	2019	35+	Lombardy region (Northen Italy)	NS	Monitoring	-Epidemiology rates -Drug consumption rates	-Level of polypharmacy (n%) -Level of comorbidities (n%) -Drug consumption (DDD)	Primary outcome	NS
Focà E, et al. [24]	2019	65+	Northern and Central Italy	HIV	Monitoring	-Epidemiology rates -Drug consumption rates -Clinical outcomes	-Level of polypharmacy (n) -Effectiveness rates (n)	Secondary outcome	NS
Grion AM, et al. [25]	2016	65+	Veneto region (Northen Italy)	NS	Decision support tools	-Epidemiology rates -Drug consumption rates	-Level of polypharmacy (n, median (IQR)) -Level of comorbidities (n, median (IQR)) -Level of PIMs (n, median (IQR)) -Level of DDI (n, median (IQR))	Primary outcome	-STOPP criteria -Micromedex
Dal Negro RW, et al. [26]	2016	18+	Veneto region (Northen Italy)	Asthma	Monitoring	-Epidemiology rates -Drug consumption rates -Costs	-Level of polypharmacy (mean) -Level of comorbidities (mean) -Direct and indirect costs (EUR/patient/year)	Secondary outcome	NS
Foo J, et al. [27]	2016	40+	Italy	COPD	Monitoring	-Epidemiology rates -Clinical outcomes -Costs	-Level of comorbidities (n) -Level of medication adherence (Morisky MMAS-8) -Effectiveness rates (%) -Direct, indirect, and societal costs (USD/patient/year)	Secondary outcome	NS

Abbreviations: COPD assessment test (CAT); chronic obstructive pulmonary disease (COPD); defined daily dose (DDD); emergency room (ER); general practitioners (GPs); lower gastrointestinal bleeding (LGIB); medical research council (mMRC) Dyspnea Scale; medication appropriateness index (MAI); multidimensional prognostic index (MPI); multisource comorbidity score (MCS); Medication Adherence Scale (MMAS); potentially inappropriate medications (PIMs); screening tool of older persons’ prescriptions (STOPP).

**Table 4 jcm-13-03529-t004:** Characteristics of included experimental studies.

Authors	Year	Study Design	Population Age	Geographical Area	Intervention Type	Tool Used	Intervention Setting (HCPs Involved)	Study Outcomes	Outcome Measures	Polypharmacy
Ghibelli S, et al. [28]	2013	Two-phase observational and experimental study	65+	Lombardy region (Northen Italy)	Decision Support Tools	INTERCHECK—CPSS	Hospital (clinical pharmacists)	-Epidemiology rates -Drug consumption rates -Clinical outcomes	Changes in parameters pre- and post-intervention: -Level of PIMs (n%) -Level of DDIs (prevalence (n%)), -Level of comorbidities (mean or %) -Drug consumption (mean or %) -Effectiveness rates (mean or %)	Secondary outcome
Onder G, et al. [29]	2008	Experimental study	65+	Lombardy region (Northen Italy)	Medication Adherence Support (case managers) and medication review	AgeD in HOme Care project	Homecare (nurses)	-Epidemiology rates	-Effectiveness rates (changes in prevalence pre- and post-intervention (n%))	Secondary outcome

Abbreviations: computerized prescription support system (CPSS); drug–drug interactions (DDIs); healthcare professionals (HCPs); potentially inappropriate medications (PIMs).

## 4. Discussion

This review highlights that there is still scarce evidence in Italy about effective interventions to monitor/manage PIP. The relationship between polypharmacy and frailty in older patients has been confirmed in both the European and Italian contexts, though interventions for the management of and improvement in these patients are still few compared with the magnitude of the problem. In this study, the most widely used intervention was monitoring, which still needs uniformity. Indeed, there is evidence that the uptake of available strategies is particularly limited in the Italian setting. This study aimed at addressing a clear need to prioritize research to generate evidence for effective interventions in patients, thus addressing an unmet need in developing patient-tailored strategies for reducing health-system burden.

Specifically, this review identified only nine studies (seven observational and two experimental) addressing HCPs to monitor and manage patients with polypharmacy carried out in the Italian healthcare setting [21,22,23,24,25,26,27,28,29]. While there are numerous studies exploring polypharmacy management strategies at the patient level, there remains a paucity of information regarding the successful implementation of frameworks for managing polypharmacy at the healthcare organizational level [30]. It is also worth noting that only two out of the nine studies conducted in Italy (22%) considered polypharmacy the primary outcome, as well as the heterogeneity of the results. Such a significant heterogeneity underscores the need to identify a commonly agreed path in order to define a gold standard. Furthermore, current practice varies considerably internationally. To back it up, various studies across different settings have explored qualitative approaches to polypharmacy management. Altogether, multidisciplinary teams (involving physicians, nurses, and pharmacists) were considered to play a significant role in optimizing prescriptions in this framework, and this was reflected in the studies; the majority of interventions involved multidisciplinary teamwork [31,32,33,34]. Although they perceived shared decision making as important, HCPs reported problems in incorporating patient prognoses and continuity of care [32,34]. 

Furthermore, the geographical distribution of the studies included in this review appears to be shifted toward the north-central regions of Italy. Indeed, only three studies encompass the entire Italian territory [21,23,27], with one of these being a European-level study that included Italy [27]. The remaining six studies have concentrated on specific areas: one on the central–northern regions [24] and three on northern Italy [22,25,26,28,29], including Lombardy [22,28,29] and Veneto [25,26]. As a result, there is a lack of research that addresses local issues and challenges regarding the topic in southern Italian regions. This research gap impairs our comprehension and capacity to address the socioeconomic dynamics of polypharmacy management in these regions.

Moreover, in the European context, Kurczewska-Michalak M, et al. (2021) have shown that the current scientific literature devotes a lot of attention to polypharmacy, not only in its general aspect, but particularly focusing on older adults [11]. Similarly, at the European level, the presence of heterogeneity emerged as a notable concern. Various potentially useful approaches to polypharmacy management have been described in the literature, although lacking a gold standard [35]. Indeed, there is evidence that the uptake of available strategies is particularly limited [34]. In addition, Kardas P. et al. (2023), part of the SIMPHATY consortium, have introduced an innovative online benchmarking application as a valuable solution in aiding clinicians and policy makers in the implementation and expansion of polypharmacy management programs for older adults [36].

Another finding of this study lies in identifying that all of the populations with polypharmacy in the studies were composed of older individuals. This underscores a notable correlation between the prevalence of polypharmacy and the coexistence of multimorbidity, thereby indicating a connection to frailty and advancing age, particularly among those aged over 65. Consistently, many studies considered polypharmacy a global risk factor among older adults [37,38], not only to monitor the occurrence of adverse effects potentially associated with polypharmacy [39,40], but also to reduce the risk of poor treatment adherence and missed doses among the geriatric population [41]. Moreover, it was found that the causal relationship between frailty and polypharmacy is unclear and, in fact, appears to be bidirectional [42]. 

Furthermore, three studies delved into specific frail patient profiles, and two of them identified that levels of polypharmacy and comorbidities were critical variables significantly influencing patient frailty. Radaelli F, et al. specifically analyzed patients with lower gastrointestinal bleeding (LGIB) and found that age, concomitant diseases, and inpatient status were variables with a critical impact on LGIB mortality risk, as reported in previous studies [21]. Furthermore, Faverio P, et al. analyzed patients with pneumonia and found older age, a higher burden of comorbidities, and a longer length of stay during the first hospitalization for pneumonia to be among the main risk factors for early rehospitalization [22]. These findings underscore the imperative for robust consideration of such factors in patient management.

Out of all of the studies included in this review, only two considered cost as one of the calculated outcomes, and interestingly, both showed a decrease in total costs after intervention [26,27]. Moreover, a systematic review of the economic impact of interventions to optimize medication use in older adults with multimorbidity and polypharmacy showed that these interventions were generally associated with a reduction in medication expenditure. Cost-utility and cost-effectiveness analyses yielded incremental cost-effectiveness ratios that were generally within the willingness-to-pay thresholds of the countries in which the studies were conducted [43].

Within the scope of this paper, it is pertinent to examine both its strengths and limitations. Accordingly, the main strength of this review is to report for the first time a summary of the Italian results of interventions to manage polypharmacy in frailty patients. To minimize bias, two review authors independently screened titles and abstracts, assessed studies for eligibility, evaluated the risk of bias, and extracted data. Additionally, employing a systematic review approach ensures that every step of the methodology remains transparent and replicable.

This review has several limitations that deserve consideration. First of all, it was limited to English-language publications, and thus, articles published in other languages were excluded. The evidence’s key limitation stemmed from the diversity among outcome measures, which varied significantly in their definitions, collection methods, and analytical approaches. This variance prevented the feasibility of conducting a meta-analysis. Furthermore, there is potential for underestimation in identifying studies due to the exclusion of grey literature.

## 5. Conclusions

This review outlines that in Italy, evidence in the literature on interventions to monitor/manage potentially inappropriate polypharmacy is limited, and none of the various potentially useful interventions are generally accepted as a “golden standard”. Given the poor implementation of polypharmacy management, it is crucial for both individual stakeholders and policy makers to encourage the use and uptake of these interventions. It becomes evident that polypharmacy is an important issue that must be considered for decision making in drug prescribing, and that it should be assessed with special caution in frail older adults. Therefore, both individual stakeholders as well as policy makers should strengthen their efforts to promote the development of patient-tailored strategies for reducing health-system burden. Further research is needed to confirm the possible benefits of reducing inappropriate polypharmacy in the development, reversion, or delay of frailty.

## Figures and Tables

**Figure 1 jcm-13-03529-f001:**
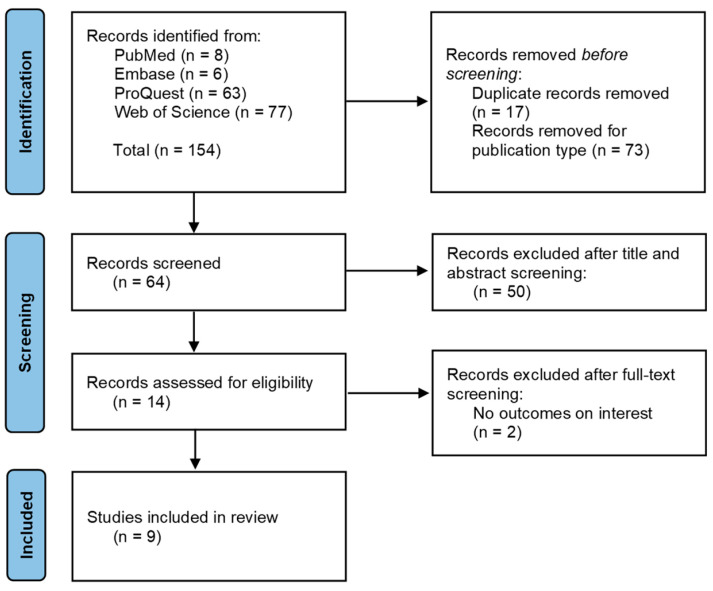
PRISMA flow chart of the literature search and study selection. (From: Page et al. [15]).

**Figure 2 jcm-13-03529-f002:**
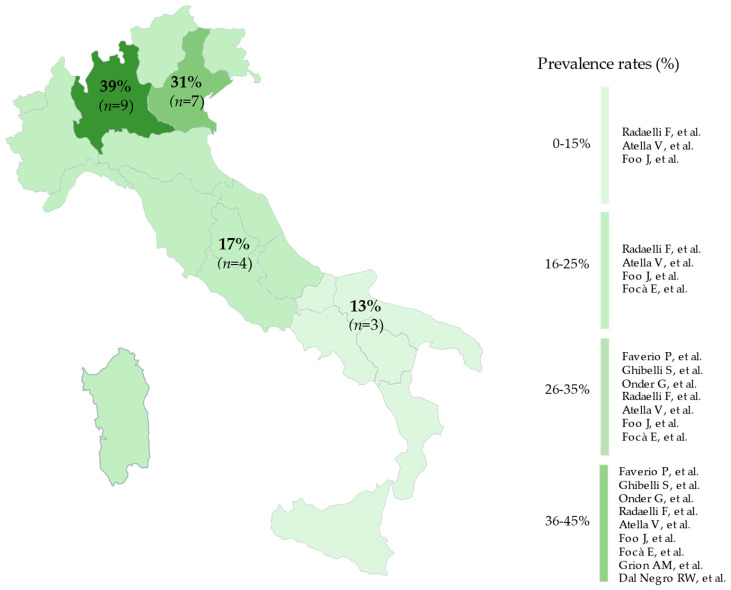
Prevalence rates of polypharmacy management interventions stratified by geographical distribution [21,22,23,24,25,26,27,28,29].

**Table 1 jcm-13-03529-t001:** Eligibility criteria.

Inclusion Criteria	Exclusion Criteria
**Study Design**	
Articles, articles in press, case studies, evaluation studies, evidence-based healthcare, comparative studies, historical articles, journal articles, pre-print papers, working papers	Books, book reviews, commentaries, conference proceedings, rationale and/or design protocols, corrections/retractions, dissertations/theses, editorials, literature reviews, letters, periodicals, conference abstracts, reviews, study protocols, consensus
**Setting**	
Italy	All other countries
**Language**	
English, Italian	All other languages
**Time Frame**	
All	NA

**Table 2 jcm-13-03529-t002:** Selection and data processing.

Data Extraction	Description
Reference	All paper identification details.
Publication year	Year of publication of paper.
Study design	Type of study conducted.
Population (P)	Adult population receiving any specific and non-specific potentially inappropriate polypharmacy.
Intervention (I)	Any type of intervention aimed at health professionals (i.e., physicians, clinicians, pharmacists, nurses), structures (i.e., nursing homes, hospitals, pharmacies), and patients, or any monitoring of potentially inappropriate polypharmacy. Intervention type and setting were considered.
Comparator (C)	No intervention or any other intervention and no monitoring activities.
Outcome (O)	Describing the effects of management of polypharmacy in terms of intervention effectiveness and cost variation.

## Data Availability

Data supporting the findings of this study are available from the corresponding author upon reasonable request.

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
