# Peer review of "Polypharmacy Management in Chronic Conditions: A Systematic Literature Review of Italian Interventions"

_jcm, 2024, doi:10.3390/jcm13123529_

Round 1

Reviewer 1 Report

Comments and Suggestions for Authors

Thank you for the opportunity to review this paper. The systematic review is on an important topic however I am unsure why the authors have decided to focus on only Italian interventions, this decision requires clarification. My detailed comments are below. 

Introduction

Line 46: Please either delete the reference to polypharmacy being more prevalent in southern Italy or explain why this is important in the context of this study. 

Please explain why looking at Italy specifically rather than a broader geographical area is warranted. Also how looking at Italy only will be beneficial to both those who work in the Italian healthcare system and those who work outside of Italy. 

Methods

PubMed is a search engine and not a database please clarify which databases you searched. I assume you mean Medline instead of PubMed. 

I would move the search strategy to an appendices.

It concerns me that 'Italy or Italian' has been included as a search terms. As I would think at lease some of papers would not consider referencing in their title, abstract or key words the country they are based in especially if study recruitment occurred across multiple countries. Please can the authors clarify that this has been considered and is not an issue. 

Results

Please include a summary of the risk of bias in the main paper. 

Discussion 

A number of European level papers are referenced it is not clear why these cannot be used in the Italian. 

The paper infers that there were two screeners please clarify. 

Eligibility criteria: Please find your definition of polypharmacy. It also seems odd to exclude papers written in Italian as this research focuses on Italy, please explain your decision. Please clarify what would happen if a paper contained data that was only partially from Italy. 

Line 93: Change 'will' to 'were' 

Comments on the Quality of English Language

Some very minor points highlighted. The paper is written in American English I assume this is correct. 

Reviewer 2 Report

Comments and Suggestions for Authors

This is a well conducted SLR (with the exception of not including Italian articles for an Italian-specific problem). The scarcity of research is highlighted.

Introduction: this SLR pertains to Italy however I struggle to identify the size of the problem in Italy - i.e. any financial implications

Methods: Why were no Italian language articles used in a SLR focusing on an issue specific to Italy? You mention that as "fundamental" to the eligibility but why?

Extensive research strategy (albeit the point above that is either a limitation (stated but needs further) or warrants repeat search) however no use of truncations - any reason in particular?

Roles need to be identified in methodology? How did scoring take place? What if there was any area of disagreement?

Where are supplementary tables 4 and 5.

Line 64: Whole sentence needs restructuring

Line 93: Wrong tense used

Line 24 page 1: should read "following abstract and full-text screenings"

Line 217: restructuring of sentence please

Limitations: I believe you mention that the outcomes were too diverse (in terms of studies included). Hence there is a significant rate of heterogeneity within this SLR. In addition, it is important to state which ones of the studies you have included was actually looking into polypharmacy as an outcome primarily and where this was just a report of background factors/demographics.

Despite some strong points, there are a couple of areas that need to be touched upon, and become more evident: are there any studies outside Italy that can be emulated in Italy? Are there any tools/interventions that are specific to Italians or are generic interventions approved/shown to be working in Italy? What are the next steps? As a report of an SLR, this is a strong piece of work - nonetheless what does this connect to in terms of Italy/Other regions/Polypharmacy in general?

Comments on the Quality of English Language

Requires some improvement - ensure the flow is appropriate and one tense is used throughout.

Round 2

Reviewer 2 Report

Comments and Suggestions for Authors

I would like to thank the authors for taking in consideration all the points raised. However, I am still struggling to understand the following three issues:

a) Why were Italian manuscripts not considered?

b) Was Pubmed used or not? (Abstract says so but main text does not).

c) Limitations: that section needs to be far larger explaining the implications of limitations; it is very important that most studies had polypharmacy as a secondary outcome and that heterogeneity was significant.

Otherwise this represents a decent effort in unearthing certain issues, however it is important that limitations are better addressed, and a review of the language used takes place.

Comments on the Quality of English Language

Please see above.
